# Vision Graph Prompting via Semantic Low-Rank Decomposition

Zixiang Ai [1]   Zichen Liu [1]   Jiahuan Zhou * [1]

## Abstract

Vision GNN (ViG) demonstrates superior performance by representing images as graph structures, providing a more natural way to capture irregular semantic patterns beyond traditional grid or sequence-based representations. To efficiently adapt ViG to downstream tasks, parameter-efficient fine-tuning techniques like visual prompting become increasingly essential. However, existing prompting methods are primarily designed for Transformer-based models, neglecting the rich topological relationships among nodes and edges in graph-based representations, limiting their capacity to model complex semantics. In this paper, we propose Vision Graph Prompting (VGP), a novel framework tailored for vision graph structures. Our core insight reveals that semantically connected components in the graph exhibit low-rank properties. Building on this observation, we introduce a semantic low-rank prompting method that decomposes low-rank semantic features and integrates them with prompts on vision graph topologies, capturing both global structural patterns and fine-grained semantic dependencies. Extensive experiments demonstrate our method significantly improves ViG's transfer performance on diverse downstream tasks, achieving results comparable to full fine-tuning while maintaining parameter efficiency. Our code is available at https://github.com/zhoujiahuan1991/ICML2025-VGP.

## 1. Introduction

Recent advent of Vision GNN (Han et al., 2022; Munir et al., 2023; 2024) has unlocked new potential by representing image patches as a graph structure, facilitating the application of GNN to diverse vision tasks. In contrast to the fixed

[1]Wangxuan Institute of Computer Technology, Peking University, Beijing, China. Correspondence to: Jiahuan Zhou <jiahuanzhou@pku.edu.cn>.

*Proceedings of the 42nd International Conference on Machine Learning*, Vancouver, Canada. PMLR 267, 2025. Copyright 2025 by the author(s).

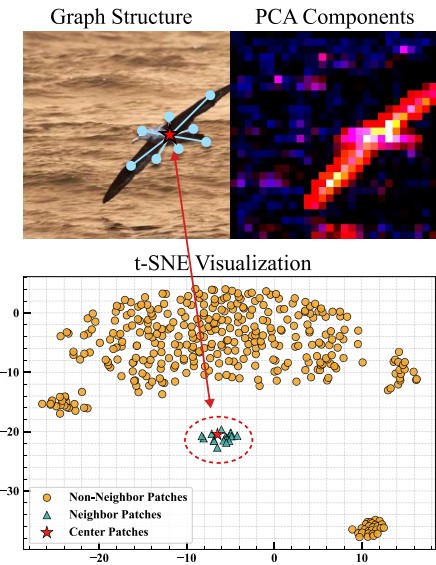

*Figure 1.* **Visualization of ViG graph structures using PCA and t-SNE.** This figure illustrates the graph structures in ViG, where semantically related components exhibit shared principal features and form a single compact cluster in t-SNE embeddings. The visualization underscores the low-rank nature of semantic information.

grid-based representations in CNNs (LeCun et al., 1998; Krizhevsky et al., 2017) or the sequential tokenization in ViTs (Dosovitskiy, 2020; Liu et al., 2021), the graph structure employed by ViG offers a more natural and flexible approach to modeling semantic patterns within images. In real-world scenarios, semantically related object parts are often distributed irregularly rather than following strict grid-like arrangements. By leveraging graph representations, ViG effectively captures these global interactions, preserving rich semantic information that is difficult to encode in traditional architectures.

When transferring pre-trained Vision GNN (ViG) models to downstream tasks, full fine-tuning is a commonly used adaptation strategy. However, as model sizes continue to increase, this method incurs substantial storage and computational overhead. To this end, parameter-efficient fine-tuning techniques, such as visual prompting (Jia et al., 2022; Bahng et al., 2022; Han et al., 2023a; Yao et al., 2025), have emerged as a promising approach, exhibiting competitive performance with significantly fewer trainable parameters. Despite previous success, existing vision prompting

methods are predominantly designed for Transformer-based architectures, struggling to adapt ViG models. Directly transferring these methods leads to suboptimal results compared to full fine-tuning, due to overlooking rich semantic information embedded in the topological structures within vision graph. Furthermore, current graph prompting techniques (Liu et al., 2023; Fang et al., 2023) are mainly developed for applications in social networks and chemical data, unable to capture the unique semantic features of visual images, thereby limiting their effectiveness in downstream vision tasks. This highlights the need for a visual prompting method that effectively captures visual semantics within vision graphs.

To address this challenge, we propose Vision Graph Prompting, a novel approach designed to capture semantic features within vision graph structures. Our method incorporates a low-rank decomposition in the prompts, grounded in the insight that the semantic information within vision graphs primarily resides in the low-rank components of the latent feature space, depicted in Figure 1. This low-rank design effectively preserves global semantic information while minimizing interference from local details. To adapt to the topological nature of vision graphs and efficiently capture relevant features, we introduce three prompt components with varying levels of granularity. First, to capture global semantic dependencies, we introduce Semantic Low-Rank Graph (SeLo-Graph) Prompt by appending trainable virtual nodes and dynamically forming edges with existing nodes, interacting with the original graph. Second, to facilitate the propagation of semantic features between connected nodes, we design Semantic Low-Rank Edge (SeLo-Edge) Prompt, incorporating the low-rank decomposition to filter out residual local details. Finally, to enhance the local fine-grained semantic features, we incorporate Semantic Low-Rank Node (SeLo-Node) Prompt that preserves intrinsic details while intensifying the low-rank semantic information. We conduct a series of experiments on downstream vision tasks to validate the superior performance of our approach. Our method significantly enhances ViG's transfer capability on downstream tasks, achieving results comparable to full fine-tuning while maintaining parameter efficiency. Our contributions can be summarized as follows:

- We introduce Vision Graph Prompting, a novel visual prompting method designed to capture semantic features within vision graph structures, overcoming the limitations of existing approaches.

- We provide a critical insight that semantic information in vision graphs primarily resides in the low-rank component of the latent feature space, leading to the development of a semantic low-rank prompt design.

- Through extensive experiments on various downstream tasks, we demonstrate that our method outperforms

existing visual prompting techniques, offering high parameter efficiency while achieving results comparable to full fine-tuning.

## 2. Related Work

### 2.1. Vision Graph Neural Network

Graph Neural Network (GNN), a critical branch of deep learning, is traditionally designed for processing graph-structured data (Kipf & Welling, 2016b; Li et al., 2019), including chemical structures, social networks, and citation networks. In computer vision, GNNs have been employed in point cloud classification and segmentation tasks by DGCNN (Wang et al., 2019) and Point-GNN (Shi & Rajkumar, 2020), as well as in human pose and action recognition (Yan et al., 2018). However, these applications are limited to tasks that can be explicitly represented as graphs.

The advent of Vision GNN (ViG) (Han et al., 2022) marked the first instance of GNN serving as a general-purpose vision backbone. ViG segments images into patches and dynamically connects them using the K-nearest neighbors algorithm. By leveraging the flexibility of graph structures to model irregular shapes, ViG surpasses grid-based (ResNet) and sequence-based (ViT) models on the ImageNet-1k (Deng et al., 2009) benchmark. Subsequent advancements like Vision HyperGraph Neural Networks (ViHGNN) (Han et al., 2023b) further enhanced ViG by employing hypergraphs to eliminate the constraint of pairwise node connections. MobileViG (Munir et al., 2023) and GreedyViG (Munir et al., 2024) improved computational efficiency, reducing graph construction overhead while preserving representational power. Despite these advances, adapting ViG-based models to downstream tasks still primarily depends on full fine-tuning, which is inefficient in terms of both parameter usage and storage. This limitation underscores the importance of exploring parameter-efficient fine-tuning (PEFT) strategies, such as visual prompting, to enhance adaptability and efficiency.

### 2.2. Visual Prompting

Prompting techniques (Brown et al., 2020; Schick & Schütze, 2020) originated in the field of natural language processing (NLP), where prompts align downstream tasks with pre-training objectives by prepending prefix words to input sequences. In computer vision, prompting has been adapted as a parameter-efficient fine-tuning technique, known as visual prompting, to customize pre-trained vision models for new tasks. Existing visual prompting methods (Li & Zhou, 2025; Liu et al., 2025) primarily target Transformer-based architectures, operating in either the input data space or latent feature space. VP (Bahng et al., 2022) learns a single prompt applied uniformly across sam-

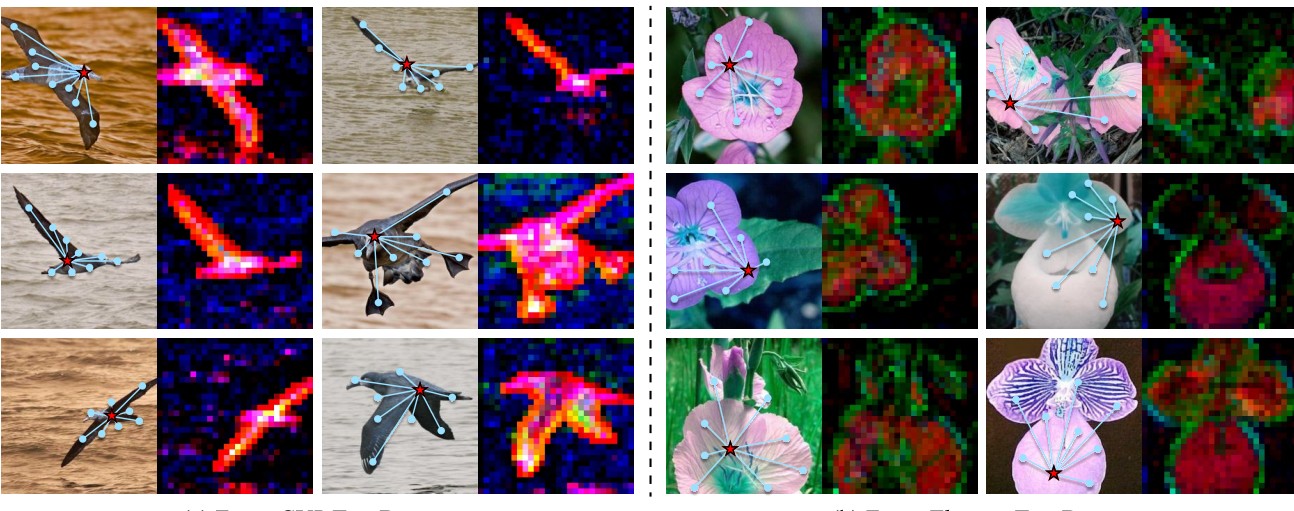

(a) From *CUB* Test Dataset          (b) From *Flowers* Test Dataset

*Figure 2.* **Observation on ViG graph structures and principal components.** The center patch (red star) is randomly selected from the primary semantic region, with neighboring patches (blue dot) linked via edges. PCA is applied to patches from the same image group (a, b), mapping the first three components to the RGB color channels. Despite variations in shape, texture, and color, the ViG graph effectively connects semantically related object parts. These connected regions share common principal components, demonstrating a low-rank structure. Background regions are filtered out by thresholding the first principal component.

ples, implemented as padding around the input image before feeding it into the pre-trained model. VPT (Jia et al., 2022) extends this approach by adding learnable prompt tokens shared across all samples, integrated within the multi-head self-attention layers of Vision Transformers. Efficiency-driven variants, such as E2VPT (Han et al., 2023a), prune prompts at both token and segment levels to mitigate the influence of redundant prompts while minimizing additional parameters. More advanced methods aim for instance-level adaptability. DAM-VP (Huang et al., 2023) addresses class-level variation by clustering samples and learning separate prompts for each cluster. InsVP (Liu et al., 2024) employs lightweight prompters that generate unique prompts per instance, enabling fine-grained adaptation. These methods have achieved performance on par with, or even surpass full fine-tuning while maintaining high parameter efficiency.

Despite these advances, applying existing visual prompting techniques to ViG models has yielded trivial results. Transformer-centric methods overlook the rich topological relationships inherent in graph-based representations, failing to exploit the structured semantic information encoded in nodes and edges. This gap highlights the need for a prompting strategy that effectively harnesses the unique characteristics of vision graphs for downstream tasks.

## 3. Observation and Insight

In this work, we propose a visual prompting approach designed to capture the semantic information embedded in vision graph structures effectively, ensuring competitive

performance compared to full fine-tuning while reducing parameter-tuning costs. To investigate the relationship between vision graph structures and semantic information, we conduct a series of experiments, as illustrated in Figure 2. By visualizing the dynamically constructed vision graph structures and analyzing the corresponding latent features using Principal Component Analysis (PCA), we gain deeper insights into the mechanisms of ViG models. This leads to our key observation:

*Semantically connected vision graph nodes, with diverse local intrinsic properties, share common PCA components, suggesting that the semantic information within vision graphs predominantly resides in the low-rank components of the latent feature space.*

Since the dominant PCA components encode the most significant variance in feature distribution, this alignment implies that semantic representations are largely concentrated in a subspace of reduced rank. Further details and additional analysis are provided in the Appendix.

## 4. Vision Graph Prompting

In this section, we detail our method for efficiently fine-tuning pre-trained ViG models, ensuring seamless integration with the original ViG workflow. To capture the low-rank semantic information embedded in the topology of vision graphs, we introduce three types of prompts: SeLo-Graph Prompt, SeLo-Edge Prompt, and SeLo-Node Prompt, each incorporating a semantic low-rank decomposition design. Given a pre-trained ViG model, we freeze the parameters

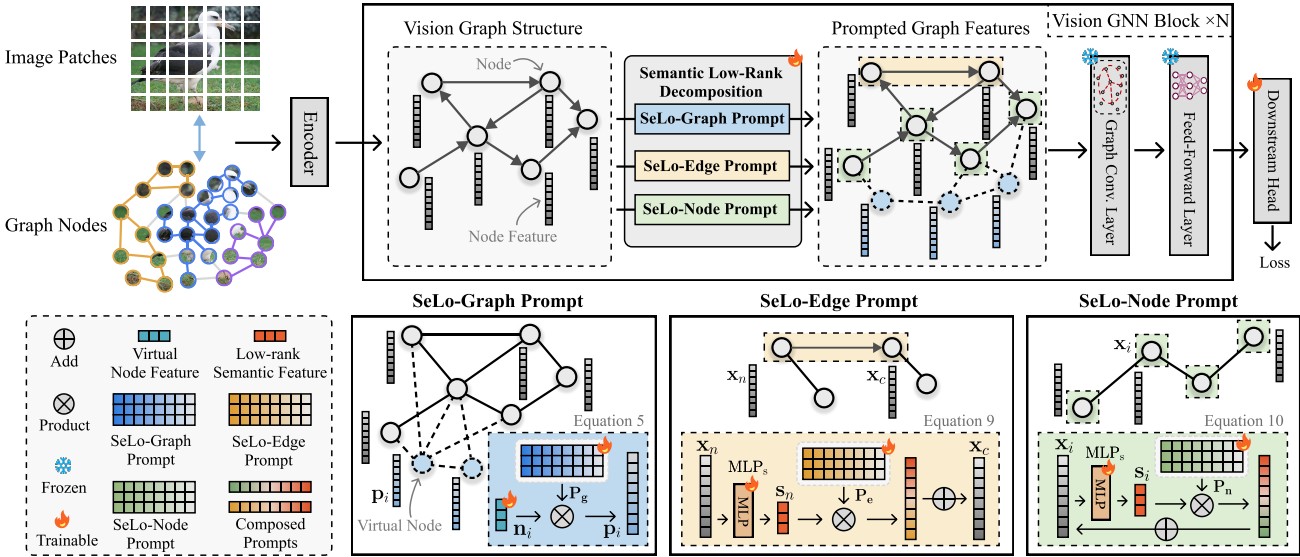

*Figure 3.* **Pipeline of Vision Graph Prompting via semantic low-rank decomposition.** Specifically, we design: (i) SeLo-Graph Prompt as trainable virtual nodes that dynamically form edges with existing nodes to capture global semantic dependencies, (ii) SeLo-Edge Prompt with low-rank decomposition to enhance feature propagation between semantically connected nodes, and (iii) SeLo-Node Prompt to refine fine-grained semantic information while preserving local intrinsic details.

of the backbone, and only the prompts along with a downstream head are fine-tuned. This approach minimizes the computational cost and parameter overhead while enabling effective adaptation to downstream tasks.

### 4.1. Preliminaries for ViG

Given an image with the size of $\mathbb{R}^{h \times w \times 3}$, ViG divides it into $N$ patches. By encoding each patch into a feature vector $\mathbf{x}_i \in \mathbb{R}^d$, we transform the image into a set of features $\mathbf{X} = [\mathbf{x}_1, \mathbf{x}_2, \ldots, \mathbf{x}_N]$, where $d$ is the feature dimension. These features are viewed as a graph of unordered nodes which are denoted as $\mathcal{V} = \{v_1, v_2, \ldots, v_N\}$. For each node $v_i$, we find its $K$ nearest neighbors $\mathcal{N}(v_i)$ and add an edge $e_{ji}$ directed from $v_j$ to $v_i$ for all $v_j \in \mathcal{N}(v_i)$. Then we obtain a graph $\mathcal{G} = (\mathcal{V}, \mathcal{E})$ where $\mathcal{E}$ denote all the edges.

By viewing an image as graph data, ViG brings GNN models into vision tasks for effective feature extraction. The image graph is processed by a stack of blocks constructed by a graph convolutional layer and a feed-forward layer. Graph convolution operates as follows:

$$\mathbf{x}_i' = f(\mathbf{x}_i, g(\mathbf{x}_i, \mathcal{N}(\mathbf{x}_i), \mathbf{W}_{agg}), \mathbf{W}_{update}), \quad (1)$$

where $\mathbf{x}_i'$ denotes updated node feature, $\mathcal{N}(\mathbf{x}_i)$ is the set of neighbor nodes of $\mathbf{x}_i$, $g(\cdot)$ and $f(\cdot)$ denote the aggregation and update operations respectively with the learnable weights $\mathbf{W}_{agg}$ and $\mathbf{W}_{update}$. Taking max-relative graph convolution for simplicity and efficiency, the process can be detailed as:

$$g(\mathbf{x}_i) = [\mathbf{x}_i, \max(\{\mathbf{x}_j - \mathbf{x}_i \mid \mathbf{x}_j \in \mathcal{N}(\mathbf{x}_i)\})] \cdot \mathbf{W}_{agg}, \quad (2)$$

$$f(\mathbf{x}_i) = \mathbf{x}_i + g(\mathbf{x}_i) \cdot \mathbf{W}_{update}. \quad (3)$$

Following that, the feed-forward layer is a simple multi-layer perceptron with two fully connected layers:

$$\mathbf{X}' = \sigma(\mathbf{X} \cdot \mathbf{W}_1) \cdot \mathbf{W}_2 + \mathbf{X}, \quad (4)$$

where $\mathbf{X}, \mathbf{X}' \in \mathbb{R}^{N \times d}$ represent the graph node features, $\mathbf{W}_1$ and $\mathbf{W}_2$ are the weights of fully-connected layers. And $\sigma$ denotes the activation function. Note that the bias term is omitted. As graph convolution layers can exchange information between nodes by aggregating features from neighbor nodes, feed-forward layers further encourage the feature transformation capacity and relieve the over-smoothing phenomenon.

### 4.2. SeLo-Graph Prompt

The vision graph consists of feature nodes and topological edges, where nodes correspond to local image patches, and edges capture semantic relationships between local regions. To effectively model global semantic dependencies within vision graph structures, we propose SeLo-Graph Prompt, a set of trainable virtual nodes that dynamically establish edges with existing nodes, enhancing long-range semantic interactions and enriching the contextual understanding of the graph.

Given a encoded vision graph $\mathcal{G} = (\mathcal{V}, \mathcal{E})$ with node features $\mathbf{X} = [\mathbf{x}_1, \mathbf{x}_2, \ldots, \mathbf{x}_N] \in \mathbb{R}^{N \times d}$, introduce a set of low-rank prompt nodes $[\mathbf{n}_1, \mathbf{n}_2, \ldots, \mathbf{n}_M] \in \mathbb{R}^{M \times r}$, where $M$ denotes the number of prompts and $r$ is the rank dimension,

chosen such that $r \ll d$. To align with the feature space of the original graph, we compose these prompts with low-rank graph-level prompt $\mathbf{P_g} \in \mathbb{R}^{r \times d}$, described as:

$$[\mathbf{p}_1, \mathbf{p}_2, \ldots, \mathbf{p}_M] = [\mathbf{n}_1, \mathbf{n}_2, \ldots, \mathbf{n}_M] \cdot \mathbf{P_g} \in \mathbb{R}^{M \times d}. \quad (5)$$

Next, we compute the cosine similarity between each prompt node $\mathbf{p}_i \in \mathbb{R}_d$ and the feature nodes $\mathbf{x}_j \in \mathbb{R}_d$, dynamically constructing a neighborhood $\mathcal{N}(\mathbf{p}_i)$ using the K nearest neighbor selection:

$$\mathcal{N}(\mathbf{p}_i) = \{\mathbf{x}_j \in \mathbf{X} \mid \mathbf{x}_j \in \text{topK}(\mathbf{p}_i \cdot \mathbf{x}_j^\top)\}. \quad (6)$$

By treating the prompts as additional virtual nodes $\overline{\mathcal{V}} = \{v_{N+1}, \ldots, v_{N+M}\}$ and connecting them to their neighbors, the constructed edges are:

$$\overline{\mathcal{E}} = \{e_{ji} \mid v_j \in \mathcal{N}(v_i), v_i \in \overline{\mathcal{V}}\}. \quad (7)$$

The updated graph $\mathcal{G}' = (\mathcal{V} \cup \overline{\mathcal{V}}, \mathcal{E} \cup \overline{\mathcal{E}})$ integrates these prompt nodes and edges, enabling richer global semantic interactions. By embedding semantic-aware prompts into the vision graph topology, this design strengthens the model's transfer capability, capturing complex relationships while maintaining parameter efficiency.

### 4.3. SeLo-Edge Prompt

In the vision graph, edges dynamically encode pairwise relationships by leveraging the similarity between node features, thereby aggregating semantically related regions within the latent feature space. As illustrated in Figure 2, these connections effectively capture semantic dependencies between object components, even in the presence of irregular shapes and varied poses, highlighting the structural coherence of the graph representation.

To leverage the contextual information embedded in these topological connections, we introduce an edge-level prompt, incorporating a semantic low-rank decomposition strategy. This design strategically filters out high-frequency local details, which often introduce noise, while promoting the propagation of semantic features from neighboring nodes to central nodes. By refining the edge-level interactions, our approach reinforces the global semantic consistency within the graph structure, leading to more robust and expressive representations for downstream tasks.

Given a center node $\mathbf{x}_c \in \mathbb{R}^d$, we define its neighborhood $\mathcal{N}(\mathbf{x}_c)$ as the set of nodes with directed edges pointing to $\mathbf{x}_c$, where each node corresponds to a feature vector. A compact semantic extraction multi-layer perceptron $\text{MLP}_s$ is employed to project each neighbor feature $\mathbf{x}_n$ into a low-rank semantic space, producing:

$$\mathbf{s}_n = \text{MLP}_s(\mathbf{x}_n) \in \mathbb{R}^r, \quad \mathbf{x}_n \in \mathcal{N}(\mathbf{x}_c), \quad (8)$$

where $\mathbf{s}_n$ denotes the semantic feature of neighbor node $\mathbf{x}_n$ and $r$ is the reduced rank dimension with $r \ll d$.

To reinforce semantic coherence within the local neighborhood, we compose the semantic feature $\mathbf{s}_n$ with a low-rank edge prompt matrix $\mathbf{P_e} \in \mathbb{R}^{r \times d}$, propagating the resulting information to the center node $\mathbf{x}_c$. The propagation is formalized as:

$$\mathbf{x}_c \leftarrow \frac{\beta}{\|\mathcal{N}_s(\mathbf{x}_c)\|} \sum_{\mathbf{s}_n \in \mathcal{N}_s(\mathbf{x}_c)} \mathbf{s}_n \cdot \mathbf{P_e} + (1 - \beta) \cdot \mathbf{x}_c, \quad (9)$$

where $\mathcal{N}_s(\mathbf{x}_c)$ denotes the set of semantic features for the neighbors of $\mathbf{x}_c$) and $\beta$ is a blending factor. By applying this semantic edge propagation across all edges, the entire graph benefits from enhanced semantic consistency and richer contextual integration.

### 4.4. SeLo-Node Prompt

Each node within the vision graph corresponds to a specific image patch, encapsulating diverse attributes such as texture, color, shape, and semantic relationships to other regions of the image. These node features inherently combine two essential aspects: intrinsic components that capture localized details and semantic components that convey the node's contextual relevance within the broader graph structure.

To better exploit this inherent duality, we propose a node-level prompt that explicitly decouples the intrinsic and semantic components via semantic low-rank decomposition, enabling finer control over the interaction between local and global information. This design ensures that the intrinsic details are preserved while amplifying the semantic consistency across the vision graph, thereby improving the overall representational capacity for downstream tasks.

Given the feature representation of a node $\mathbf{x}_i \in \mathbb{R}^d$, we decompose the low-rank semantic components $\mathbf{s}_i \in \mathbb{R}^r$ through the semantic extraction module $\text{MLP}_s$, previously utilized in the edge-level prompt. To further consolidate semantic information, we introduce a low-rank node-level prompt $\mathbf{P_n} \in \mathbb{R}^{r \times d}$, which is learned across all graph nodes to capture common features shared among different image regions. Here, $r$ and $d$ respectively denote the rank dimension and feature dimension of the ViG model, with $r \ll d$.

Then, the node feature is updated by integrating the refined semantic component back into the intrinsic feature space:

$$\mathbf{x}_i \leftarrow \alpha \cdot \mathbf{s}_i \cdot \mathbf{P_n} + (1 - \alpha) \cdot \mathbf{x}_i, \quad (10)$$

where $\alpha$ is a blending hyperparameter that balances the contribution of intrinsic and semantic components. By selectively enhancing semantic consistency while preserving critical local details, the node-level prompt enables a more expressive and robust representation of image patches, im-

*Table 1.* Head-tuning adaptation performance of various methods across ten datasets using the pyramid ViG model pre-trained on ImageNet-21k, with classification accuracy (%) reported. For reference, state-of-the-art visual prompting methods applied to ViT models pre-trained on ImageNet-21k are also included.

| Methods | Backbone | DTD | CUB | NABirds | Dogs | Flowers | Food | CIFAR | CIFAR10 | GTSRB | SVHN | **Average** |
|---|---|---|---|---|---|---|---|---|---|---|---|---|
| Full Fine-Tune | ViG-M | 73.5 | 89.0 | 82.8 | 81.4 | 98.5 | 87.4 | 89.2 | 98.6 | 98.0 | 91.7 | 89.0 |
| Full Fine-Tune | ViT-B | 64.3 | 87.3 | 82.7 | 89.4 | 98.8 | 84.9 | 68.9 | 97.4 | 97.1 | 87.4 | 85.8 |
| VPT | ViT-B | 65.8 | 88.5 | 84.2 | 90.2 | 99.0 | 83.3 | 78.8 | 96.8 | 90.7 | 78.1 | 85.5 |
| DAM-VP | ViT-B | 73.1 | 87.5 | 82.1 | 92.3 | 99.2 | 86.9 | 88.1 | 97.3 | 90.6 | 87.9 | 88.5 |
| AutoVP | ViT-B | 62.5 | 85.4 | 83.5 | 90.3 | 90.4 | 82.3 | 77.9 | 95.2 | 93.1 | 92.9 | 85.4 |
| InsVP | ViT-B | 74.5 | 89.3 | 84.6 | 93.6 | 99.2 | 89.5 | 91.3 | 98.4 | 96.1 | 96.1 | 91.3 |
| Linear | ViG-M | 66.7 | 76.2 | 71.3 | 71.3 | 81.4 | 79.2 | 67.2 | 89.4 | 77.4 | 65.5 | 74.6 |
| Adapter | ViG-M | 68.3 | 74.4 | 76.0 | 66.8 | 83.5 | 84.7 | 82.6 | 95.5 | 93.5 | 93.1 | 81.8 |
| VP | ViG-M | 67.4 | 81.1 | 74.5 | 72.0 | 94.3 | 77.4 | 78.6 | 93.1 | 89.0 | 83.6 | 81.1 |
| VPT | ViG-M | 71.4 | 77.3 | 76.4 | 73.1 | 95.3 | 81.9 | 76.3 | 93.2 | 79.7 | 82.4 | 80.7 |
| DAM-VP | ViG-M | 71.8 | 82.6 | 77.4 | 74.2 | 95.9 | 82.3 | 81.5 | 94.9 | 91.4 | 85.1 | 83.7 |
| InsVP | ViG-M | 69.8 | 85.0 | 78.0 | 77.0 | 96.1 | 84.1 | 83.3 | 95.8 | 87.6 | 89.4 | 84.6 |
| VFPT | ViG-M | 72.1 | 82.3 | 77.2 | 75.6 | 95.9 | 83.2 | 82.4 | 95.4 | 85.4 | 86.1 | 83.6 |
| GraphPrompt | ViG-M | 63.4 | 72.5 | 76.9 | 69.9 | 83.6 | 80.2 | 81.4 | 94.2 | 91.2 | 92.6 | 80.6 |
| GPF-Plus | ViG-M | 71.0 | 82.0 | 77.2 | 78.2 | 95.7 | 82.6 | 80.9 | 94.5 | 90.5 | 83.1 | 83.6 |
| **VGP**(Ours) | ViG-M | **74.8** | **87.4** | **80.9** | **81.7** | **98.2** | **89.5** | **89.7** | **98.3** | **98.1** | **96.9** | **89.6** |

proving the graph's capacity to handle complex downstream tasks.

### 4.5. Analysis and Discussion

In this section, we analyze and discuss how our method promotes the feature extraction of Vision GNNs, effectively capturing the critical semantic information.

As shown in Figure 3, the SeLo-Graph Prompt dynamically refines original graph structures. Given initial node features $\mathbf{X} = [\mathbf{x}_1, \mathbf{x}_2, \ldots, \mathbf{x}_N] \in \mathbb{R}^{N \times d}$, we extend the graph by appending virtual nodes $[\mathbf{n}_1, \mathbf{n}_2, \ldots, \mathbf{n}_M] \cdot \mathbf{P_g} \in \mathbb{R}^{M \times d}$, thus building new edges $\bar{\mathcal{E}}$ within the vision graph, as formulated in Equation 7. This structural adaptation leads to an updated neighborhood relationship, denoted as $\hat{\mathcal{N}}(\cdot)$, effectively enriching the connectivity patterns in the graph representation.

Beyond structural refinement, SeLo-Node Prompt and SeLo-Edge Prompt operate at the feature level, modulating the convolutional process rather than altering the graph topology explicitly. With SeLo-Node Prompt $\mathbf{P_n} \in \mathbb{R}^{r \times d}$ involved, the aggregation operation in Equation 2 is reformulated as:

$$\hat{g}(\mathbf{x}_i, \mathbf{P_n}) = g((1 - \alpha) \cdot \mathbf{x}_i + \alpha \cdot \mathrm{MLP_s}(\mathbf{x}_i) \cdot \mathbf{P_n}), \quad (11)$$

where $\hat{g}(\cdot)$ represents reformulated aggregation process and $g(\cdot)$ is original version in Equation 2.

Similarly, SeLo-Edge Prompt $\mathbf{P_e} \in \mathbb{R}^{r \times d}$ influences the update mechanism in graph convolution, reformulated as:

$$\hat{f}(\mathbf{x}_i, \mathbf{P_e}) = (1 - \beta) \cdot \mathbf{x}_i + \hat{g}(\mathbf{x}_i, \mathbf{P_n}) \cdot \mathbf{W}_{update}$$
$$+ \sum_{\mathbf{x}_j \in \hat{\mathcal{N}}(\mathbf{x}_i)} \beta \cdot \mathrm{MLP_s}(\mathbf{x}_j) \cdot \mathbf{P_e}, \quad (12)$$

where $\hat{f}(\cdot)$ denotes prompted update operation. From Equation 12, we can see that our semantic low-rank prompts collectively work together, facilitating both structural adaptation and feature enhancement. The SeLo-Node Prompt operates on the aggregation process while the SeLo-Edge Prompt enhances the update process of graph convolution. Meanwhile, the SeLo-Graph Prompt strengthens both processes by refining the neighborhood topology, represented by $\hat{\mathcal{N}}(\mathbf{x}_i)$. Together, these components effectively balance structural adaptation and feature enhancement, reinforcing the model's ability to extract robust semantic representations.

## 5. Experiments

We conduct extensive experiments to evaluate the efficacy of our proposed method across diverse domains. Specifically, we test on ten image classification datasets using a pre-trained Vision GNN (ViG) backbone to validate its performance on vision tasks. Additionally, to demonstrate the generalizability of our approach, we extend it to traditional graph tasks, evaluating nine datasets from the fields of chemistry and biology. The results highlight the adaptability and robustness of our method across different scenarios.

*Table 2.* Adaptation performance on eight chemistry and one biology benchmarks, based on GIN models pre-trained with Edge Prediction.

| Methods | BBBP | Tox21 | ToxCast | SIDER | ClinTox | MUV | HIV | BACE | PPI | Average |
|---|---|---|---|---|---|---|---|---|---|---|
| Full Fine-Tune | 66.56 | 78.67 | 66.29 | 64.35 | 69.07 | 79.67 | 77.44 | 80.90 | 71.54 | 72.72 |
| GPPT | 64.13 | 66.41 | 60.34 | 54.86 | 59.81 | 63.05 | 60.54 | 70.85 | 56.23 | 61.80 |
| GPPT (w/o ol) | 69.43 | 78.91 | 64.86 | 60.94 | 62.15 | 82.06 | 73.19 | 70.31 | 76.85 | 70.97 |
| GraphPrompt | 69.29 | 68.09 | 60.54 | 58.71 | 55.37 | 62.35 | 59.31 | 67.70 | 49.48 | 61.20 |
| GPF | 69.57 | 79.74 | 65.65 | 67.20 | 69.49 | 82.86 | 77.60 | 81.57 | 76.98 | 74.51 |
| GPF-Plus | 69.06 | 80.04 | 65.94 | **67.51** | 68.80 | **83.13** | 77.65 | 81.75 | 77.00 | 74.54 |
| **VGP**(Ours) | **71.53** | **80.11** | **68.53** | 65.65 | **74.21** | 82.60 | **80.69** | **83.59** | **80.58** | **76.39** |

## 5.1. Datasets

**Vision Datasets.** For vision tasks, we employ 10 benchmarks listed in Table 1, covering various categories and diverse distributions, following the approaches of DAM-VP (Huang et al., 2023) and InsVP (Liu et al., 2024). The selected datasets include *CIFAR10* (Krizhevsky et al., 2009), *CIFAR* (Krizhevsky et al., 2009), *DTD* (Cimpoi et al., 2014), *CUB* (Wah et al., 2011), *NABirds* (Van Horn et al., 2015), *Stanford Dogs* (Khosla et al., 2011), *Oxford Flowers* (Nilsback & Zisserman, 2008), *Food* (Bossard et al., 2014), *GT-SRB* (Stallkamp et al., 2012), and *SVHN* (Netzer et al., 2011). The same data augmentation strategy is adopted across all compared methods, involving randomly resizing the input images to $256 \times 256$, followed by cropping them to $224 \times 224$. More details are provided in the Appendix.

**Graph Datasets.** To further examine the extendability of the model to traditional graph tasks, we select downstream datasets from the chemistry and biology domains, following GPF-Plus (Fang et al., 2023). For the chemistry domain, we use eight graph classification datasets from MoleculeNet (Wu et al., 2018) as downstream tasks. For the biology domain, we utilize a dataset consisting of 88K labeled protein ego networks, designed for predicting 5000 coarse-grained biological functions, referred to as the PPI dataset. We adopt the challenging scaffold split for the chemistry datasets and the species split for the biology dataset, ensuring alignment with prior works (Hu et al., 2019).

## 5.2. Comparing Methods

As for vision tasks, we compare our method with both visual prompting and graph prompting methods. We also report the full fine-tuning results as a baseline. For visual prompting methods, we compare with task-level approaches such as VP (Bahng et al., 2022), VPT (Jia et al., 2022), and VFPT (Zeng et al., 2024), the cluster-level visual prompting method DAM-VP (Huang et al., 2023), as well as the latest InsVP (Liu et al., 2024). Additionally, we apply graph-specific prompting methods, including GraphPrompt (Liu et al., 2023) and GPF-Plus (Fang et al., 2023) to vision tasks for comparison. All methods were trained for 100 epochs

with consistent data augmentation strategies, in line with previous works to ensure a fair comparison.

For graph tasks, we compare with traditional graph prompting methods, including GPPT (Sun et al., 2022), GPPT without orthogonal prompt constraint loss, denoted as GPPT (w/o ol), GraphPrompt (Liu et al., 2023), GPF (Fang et al., 2022), and GPF-Plus (Fang et al., 2023). Full fine-tuning results are also reported for reference.

## 5.3. Quantitative Analysis

**Analysis on Vision Tasks.** As shown in Table 1, our method excels all the existing visual prompting techniques across ten datasets with diverse distributions, achieving an average accuracy improvement of **5.0%**. This significant gain can be attributed to our prompt designs at the graph, edge, and node levels, which effectively capture the rich semantic information embedded within the topological graph structures. It is worth noting that our approach even outperforms full fine-tuning on smaller datasets like *Food* and *SVHN* while requiring far fewer trainable parameters. This advantage comes from our lightweight low-rank decomposition, which efficiently preserves fine-grained semantic information while maintaining parameter efficiency.

Interestingly, when adapting visual prompting methods originally designed for Vision Transformers to ViG models, we observe a noticeable performance drop due to their reliance on the attention mechanism. Among these, data-level prompting methods such as InsVP mitigate the issue to some extent. Furthermore, our approach also surpasses graph-specific prompting methods adapted from traditional graph domains. This superiority stems from our key insight into the semantic coherence and low-rank properties inherent to graph-based representations, which are effectively leveraged by our semantic low-rank decomposition prompting design.

**Analysis on Graph Tasks.** To further validate the generalizability of our semantic low-rank prompts, we extend them to traditional graph tasks. As shown in Table 2, our method achieves strong performance across graph datasets, surpassing prior approaches on seven out of nine datasets

*Table 3.* Ablation study on effects of each semantic low-rank prompt. We report the classification accuracy (%) on the *CUB* and *GTSRB* datasets for evaluation.

| SeLo-Graph | SeLo-Edge | SeLo-Node | *CUB* | *GTSRB* |
|:---:|:---:|:---:|:---:|:---:|
| Linear Probing | | | 76.2 | 77.4 |
| ✓ | - | - | 81.9 | 86.9 |
| ✓ | ✓ | - | 85.3 | 93.0 |
| ✓ | ✓ | ✓ | **87.4** | **98.1** |

*Table 4.* Ablation study on rank dimension $r$. Experiments are conducted on four datasets, with trainable parameters (Param.) reported. Since the size of the downstream head varies with specific datasets, we report the Param. of *CUB* for reference.

| Rank $r$ | Param.(M) | *CUB* | *CIFAR* | *GTSRB* | *SVHN* |
|:---:|:---:|:---:|:---:|:---:|:---:|
| 4 | 1.50 | 85.8 | 89.1 | 97.6 | 96.3 |
| 8 | 1.70 | 86.2 | 89.2 | 97.7 | 96.5 |
| 16 | 2.10 | 86.7 | 89.5 | 98.0 | **96.9** |
| 32 | 2.90 | **87.4** | 89.7 | **98.1** | **96.9** |
| 64 | 4.49 | 87.2 | **89.8** | 97.9 | 96.7 |
| 128 | 7.69 | 86.5 | 89.6 | 97.8 | 96.2 |
| 384 | 20.49 | 86.1 | 89.3 | 97.6 | 95.8 |

and attaining a notable **3.67%** improvement over full fine-tuning. Although graph data in the chemistry and biology domains lacks explicit visible semantic connections, we hypothesize that low-rank patterns exist within structural relationships, such as chemical bonds or protein interactions, analogous to those in visual objects. By leveraging our low-rank decomposition design, our method effectively captures these latent relationships, yielding significant performance gains on graph-based tasks.

## 5.4. Ablation Study

**Effect of Core Components.** Our Vision Graph Prompting (VGP) integrates three key components: SeLo-Graph Prompt, SeLo-Edge Prompt, and SeLo-Node Prompt. As a parameter-efficient fine-tuning strategy, VGP freezes entire backbone parameters and optimizes only the prompts alongside a shallow downstream head. To verify the individual effectiveness of each component, we incrementally incorporate them into a linear probing setup on the ViG backbone, using *CUB* and *GTSRB* datasets for evaluation. The introduction of the SeLo-Graph Prompt yields a notable performance gain of **5.7%** and **9.5%** on the respective datasets. Subsequent incorporation of the SeLo-Edge Prompt and SeLo-Node Prompt further improves the classification accuracy, achieving the peak performance on both datasets. It verifies the efficacy of our components in capturing essential semantic information embedded in the topological structure of vision graphs, validating the rationality of our designs.

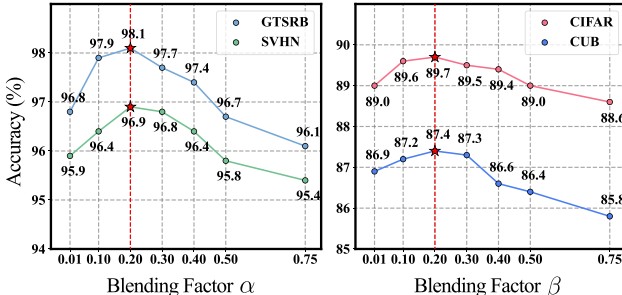

*Figure 4.* Ablation on blending factors $\alpha$ and $\beta$. The red star marks the best result.

**Ablation on Rank Dimension $r$.** In our semantic low-rank prompt design, the rank dimension $r$ is an essential hyper-parameter. As shown in Table 4, we conduct experiments on four datasets to probe the optimal $r$. The searching scope ranges from 4 to 384, where 384 matches the primary feature dimension of our ViG backbone. Results indicate that both overly small and excessively large $r$ values lead to suboptimal performance. Specifically, a very small $r$ limits the model to represent high-dimensional semantic information, while an excessively large $r$ fails to filter high-frequency local details, disrupting the extraction of low-rank semantic features. Furthermore, trainable parameters increase linearly with the $r$. To balance performance and efficiency, we set $r$ to 32.

**Ablation on Blending Factors $\alpha$ and $\beta$.** We evaluate the impact of blending factors $\alpha$ and $\beta$, which respectively control the mix-up ratio of the SeLo-Node Prompt and SeLo-Edge Prompt with the original node features. As shown in Figure 4, experiments on four datasets reveal that both $\alpha$ and $\beta$ achieve near-optimal performance within the range of 0.1 to 0.3. Deviation from this range slightly degrades performance due to an imbalance preserving local intrinsic details and amplifying semantic information. Empirically, we set $\alpha$ and $\beta$ to 0.2 for all experiments.

## 6. Conclusion

In this paper, we propose a novel visual prompting method, namely Vision Graph Prompting (VGP), with a semantic low-rank decomposition design. To the best of our knowledge, we are the first exploration on prompting ViG models. Our approach is built upon the insight that the semantic information within vision graph structures resides in the low-rank components of latent feature space, enabling both improved performance and parameter efficiency. Extensive experiments on diverse downstream tasks demonstrate that our method outperforms state-of-the-art visual and graph prompting methods, achieving results comparable to full fine-tuning while substantially reducing trainable parameters. Furthermore, the adaptation of our semantic low-rank design to traditional graph tasks highlights its broader potential, offering a promising direction for future research.

## Acknowledgments

This work was supported by the National Natural Science Foundation of China (62376011) and the National Key R&D Program of China (2024YFA1410000).

## Impact Statement

This paper presents work whose goal is to advance the field of Machine Learning. There are many potential societal consequences of our work, none of which we feel must be specifically highlighted here.

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

# A. Appendix

## A.1. Analysis on Efficiency

As shown in Table 5, we analyze the efficiency of our Vision Graph Prompting (VGP) in terms of parameter efficiency and computational cost. As a parameter-efficient fine-tuning (PEFT) approach, our method aims to reduce trainable parameters, which means much less GPU memory consumption and storage burden, while retaining comparable performance against full fine-tuning and little computational overhead.

**Parameter Efficiency.** The key contribution of our method lies in its parameter-efficient design. In contrast to full fine-tuning, where all parameters are updated, VGP only updates a small portion of prompt parameters. Attributed to our semantic low-rank design, our method reduces **94.6%** trainable parameters averagely while maintaining comparable even superior performance against full fine-tuning, demonstrated in Table 1 and Table 5 respectively.

**Computational Efficiency.** In terms of computational cost, our VGP introduces little additional overhead of textbf3.1% averagely, demonstrated in Table 5. Attributed to our semantic low-rank decomposition design, we save a lot of computation by reducing feature dimension to low-rank dimension for computing, while extracting critical semantic information and filtering out disruption of noisy local details. Our approach does not require the re-training of the entire model, which helps to mitigate the computational cost typically associated with full fine-tuning.

In summary, the VGP framework provides a highly efficient approach to adapting Vision GNNs for downstream tasks, offering significant trainable parameters reduction and neglectable computational cost, while maintaining competitive performance compared to full fine-tuning.

## A.2. Implementation Details

Our experiments on vision tasks are based on a medium pyramid Vision GNN model pre-trained on ImageNet-21k (Krizhevsky et al., 2017). With the backbone parameters frozen, only our prompt modules and the task-specific head are trained. Following DAM-VP (Huang et al., 2023), we train for 100 epochs for each dataset and incorporate 10 additional epochs for probing the optimal result. We utilize the AdamW (Loshchilov & Hutter, 2017) optimizer for optimization and implement cosine learning rate annealing. The learning rate is set as 0.001 and the weight decay is 0.05. Regarding the graph tasks, we follow the approach of GPF-Plus (Fang et al., 2023), utilizing a widely used 5-layer GIN as the underlying architecture. The GIN model is pre-trained on the chemistry and biology datasets correspondingly with the Edge Prediction strategy (Kipf & Welling, 2016a).

## A.3. Analysis on Semantic Low-Rank Property

In this work, we explore the feature extraction mechanism behind Vision GNN models. By visualizing vision graph structures, we find that semantically related object parts are consistently connected by dynamically constructed edges, regardless of variations in object shape, pose, or local texture. Additionally, we utilize Principal Component Analysis (PCA) to decompose the latent features and visualize the dominant components by mapping them to RGB color channels, as shown in Figure 2. Notably, semantically connected patches exhibit similar PCA components, suggesting an inherent low-rank property.

To provide a theoretical foundation for this observation, we recall that PCA is a widely used dimensionality reduction technique that projects high-dimensional data onto a lower-dimensional subspace via a linear transformation, maximizing variance retention.

Principal Component Analysis (PCA) is a common data dimensionality reduction method. It maps data from a high-dimensional space to a low-dimensional space through a linear transformation while preserving as much variance as possible. The essence of PCA is indeed the Eigenvalue Decomposition (EVD) based on the covariance matrix, which strictly retains rank equivalence.

Given a set of normalized node features $\mathbf{X} = [\mathbf{x}_1, \ldots, \mathbf{x}_n] \in \mathbb{R}^{N \times d}$, where $n$ is number of nodes and $d$ represents feature dimension, $N \gg d$. The corresponding covariance matrix $\Sigma$ can be computed as:

$$\mathbf{\Sigma} = \mathbf{X}^T \cdot \mathbf{X} \in \mathbb{R}^{d \times d}, \tag{13}$$

The eigenvalue decomposition of the covariance matrix can be formulated as:

$$\mathbf{\Sigma} = \mathbf{V} \cdot \mathbf{\Lambda} \cdot \mathbf{V}^T, \tag{14}$$

$$\text{where} \quad \mathbf{\Lambda} = \text{diag}(\lambda_1, \lambda_2, \ldots, \lambda_d), \quad \lambda_1 \geq \lambda_2 \geq \cdots \geq \lambda_d \geq 0 \tag{15}$$

$$\text{and} \quad \mathbf{V} = [\mathbf{v}_1, \ldots, \mathbf{v}_d] \in \mathbb{R}^{d \times d}. \tag{16}$$

Then we can express $\mathbf{\Sigma}$ equivalently as:

$$\mathbf{\Sigma} = \sum_{i=1}^{d} \lambda_i \mathbf{v_i} \cdot \mathbf{v_i}^T. \tag{17}$$

The rank dimension $r$ equals the non-zero number of eigenvalues $\lambda_i$. In practice, eigenvalues $\lambda_i$ less than a certain threshold $\epsilon$ can be regarded as zero. Assuming that the reservation is made before $r$ principal components, the error term can be estimated by $O(\lambda_{r+1})$.

As depicted in Figure 2 and Figure 5, we visualize ViG graph structures alongside their PCA component coefficients. From Figure 2, we can see that the semantic parts share similar coefficients of the first three dominant PCA components, which entails most feature distribution variance. Furthermore, we visualize the coefficient magnitude distribution of the PCA components, and an obvious long-tail phenomenon can be observed. Setting error threshold $\epsilon = 0.25$, the rank $r$ of connected patches in *CUB* and *Flowers* dataset can be estimated as $50$ and $60$ respectively, significantly lower than the original feature dimension $d = 768$. Based on this semantic low-rank observation, we design our VGP method, achieving both efficiency and efficacy in adapting ViG models.

### A.4. Details of Vision Datasets

As shown in Table 6, we provide more detailed information for our vision adaptation datasets. Diverse target classes and rich categories have been covered by these benchmarks, verifying the generalizability of our proposed method.

*Table 5.* Efficiency metrics of our method compared to full fine-tuning across ten datasets in downstream vision task. Trainable parameters (Param) and FLOPs are reported to evaluate parameter efficiency and computational costs.

| Methods | Backbone | Metric | DTD | CUB | NABirds | Dogs | Flowers | Food | CIFAR | CIFAR10 | GTSRB | SVHN | Average |
|---|---|---|---|---|---|---|---|---|---|---|---|---|---|
| Full Fine-Tune | ViG-M | Param. (M) | 48.55 | 48.71 | 49.54 | 48.62 | 48.61 | 48.6 | 48.6 | 48.51 | 48.54 | 48.51 | 48.68(100%) |
| | | FLOPs (G) | 8.94 | 8.94 | 8.94 | 8.94 | 8.94 | 8.94 | 8.94 | 8.94 | 8.94 | 8.94 | 8.94(100%) |
| **VGP**(Ours) | ViG-M | Param. (M) | 2.47 | 2.63 | 3.46 | 2.55 | 2.53 | 2.53 | 2.53 | 2.44 | 2.47 | 2.44 | 2.61(-94.6%) |
| | | FLOPs (G) | 9.22 | 9.22 | 9.22 | 9.22 | 9.22 | 9.22 | 9.22 | 9.22 | 9.22 | 9.22 | 9.22(+3.1%) |

*Table 6.* Vision dataset statistics used in our downstream adaptation tasks.

| Dataset | Target Class | Categories | Train | Val | Test |
|---|---|---|---|---|---|
| DTD (Cimpoi et al., 2014) | textures | 47 | 1,880 | 1,880 | 1,880 |
| CUB200 (Wah et al., 2011) | birds | 200 | 5,394 | 600 | 5,794 |
| NABirds (Van Horn et al., 2015) | birds | 555 | 21,536 | 2,393 | 24,633 |
| Stanford-Dogs (Khosla et al., 2011) | dogs | 120 | 10,800 | 1,200 | 8,580 |
| Oxford-Flowers (Nilsback & Zisserman, 2008) | flowers | 102 | 1,020 | 1,020 | 6,149 |
| Food101 (Bossard et al., 2014) | food dishes | 101 | 60,600 | 15,150 | 25,250 |
| CIFAR100 (Krizhevsky et al., 2009) | all | 100 | 40,000 | 10,000 | 10,000 |
| CIFAR10 (Krizhevsky et al., 2009) | all | 10 | 40,000 | 10,000 | 10,000 |
| GTSRB (Stallkamp et al., 2012) | traffic signs | 43 | 21,312 | 2,526 | 12,630 |
| SVHN (Netzer et al., 2011) | numbers | 10 | 58,605 | 14,652 | 26,032 |

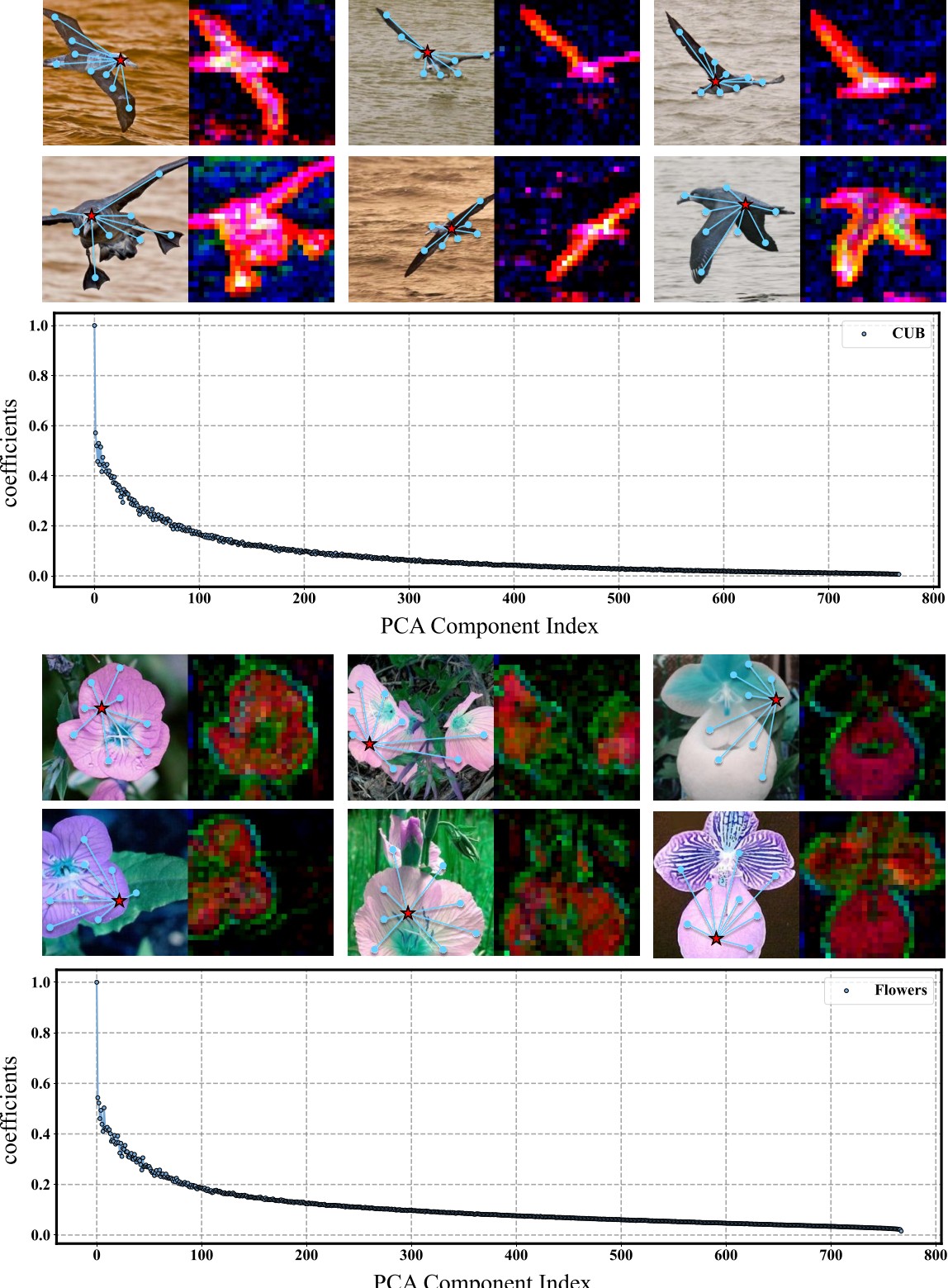

*Figure 5.* Visualization of ViG graph structures, the first three major PCA components, and the average distribution of PCA component coefficient magnitudes.

