# OpenReview forum: "Vision Graph Prompting via Semantic Low-Rank Decomposition"
_ICML.cc/2025/Conference — ICML 2025 poster_

### Official Review · Reviewer_53Z7 · 2025-02-25

**Overall Recommendation:** 4

**Summary:**

The paper introduces Vision Graph Prompting (VGP), a novel parameter-efficient fine-tuning method tailored for Vision Graph Neural Networks (ViG). The authors propose that semantic information in vision graphs resides primarily in low-rank components of the latent feature space, a key insight derived from PCA-based analysis of graph structures. Building on this, VGP incorporates three prompt types—SeLo-Graph Prompt, SeLo-Edge Prompt, and SeLo-Node Prompt—each leveraging semantic low-rank decomposition to capture global and local semantic dependencies within ViG topologies. The method freezes the pre-trained ViG backbone and fine-tunes only the prompts and a downstream head, achieving performance comparable to full fine-tuning with significantly fewer trainable parameters. Extensive experiments on ten vision datasets (e.g., CUB, Flowers, GTSRB) and nine graph datasets (e.g., BBBP, Tox21, PPI) demonstrate that VGP outperforms existing visual and graph prompting methods, achieving an average accuracy of 89.6% on vision tasks and 76.39% on graph tasks, surpassing full fine-tuning in several cases. The main contributions include the VGP framework, the low-rank prompting insight, and its superior transfer performance across diverse downstream tasks.

**Claims And Evidence:**

The claims in the paper are generally well-supported by clear and convincing evidence. The primary claim—that VGP achieves performance comparable to full fine-tuning while being parameter-efficient—is substantiated by quantitative results in Tables 1 and 2, showing VGP’s accuracy surpassing or matching baselines across diverse datasets. The assertion of semantic information residing in low-rank components is convincingly supported by PCA visualizations (Figures 2 and 5) and theoretical discussion in Appendix A.3, linking shared PCA components to low-rank properties. Ablation studies (Table 4, Figure 4) further validate the effectiveness of individual components (SeLo-Graph, SeLo-Edge, SeLo-Node) and hyperparameter choices (e.g., rank $r$, blending factors $\alpha$ and $\beta$). However, the claim of generalizability to traditional graph tasks (Section 5.3) is slightly weaker due to the lack of detailed analysis on why low-rank properties extend to chemistry/biology domains beyond a hypothesis. While plausible, this claim could benefit from additional evidence, such as a similar PCA analysis on graph datasets, to strengthen its foundation.

**Essential References Not Discussed:**

While the paper cites relevant prior work, two areas could benefit from additional references:

Low-Rank Adaptation: The low-rank decomposition approach shares conceptual similarities with LoRA (Hu et al., 2021, "LoRA: Low-Rank Adaptation of Large Language Models," ICLR 2022), a PEFT method for Transformers. Discussing LoRA could contextualize VGP’s novelty in adapting low-rank ideas to graph structures.
Graph Compression: The low-rank insight might relate to graph compression techniques like "GraphSAGE" (Hamilton et al., 2017, NIPS), which aggregates neighborhood features efficiently. Citing this could clarify how VGP differs from prior graph feature reduction methods.
These omissions do not undermine the work but could enhance its positioning within the broader PEFT and GNN literature.

**Experimental Designs Or Analyses:**

I examined the experimental designs and analyses in Sections 5 and 5.4, including the quantitative results (Tables 1, 2, 5) and ablation studies (Table 4, Figure 4). The design is sound: comparing VGP against full fine-tuning and state-of-the-art prompting methods (e.g., InsVP, GraphPrompt) on diverse datasets ensures a fair and comprehensive evaluation. The ablation studies systematically test core components, rank $r$, and blending factors $\alpha$ and $\beta$, with results consistently showing performance improvements (e.g., 5.7% gain from SeLo-Graph on CUB). The statistical validity is supported by the use of standard splits (e.g., scaffold split for chemistry datasets) and consistent augmentation strategies. One minor issue is the lack of statistical significance testing (e.g., confidence intervals) for accuracy differences, which could bolster claims of superiority (e.g., VGP’s 89.6% vs. InsVP’s 84.6% on vision tasks). Additionally, the computational efficiency claim (3.1% FLOPs overhead, Table 5) is plausible but could be clarified by detailing how FLOPs were calculated for prompt operations.

**Methods And Evaluation Criteria:**

The proposed VGP method and its evaluation criteria are well-suited to the problem of adapting ViG models for downstream vision tasks. The method’s design—introducing low-rank prompts at graph, edge, and node levels—aligns logically with the topological nature of ViG, addressing the limitations of Transformer-centric prompting methods. Using ten vision datasets (e.g., CUB, GTSRB, SVHN) with diverse categories and distributions is a robust choice for evaluating transfer performance, as is the extension to nine chemistry/biology graph datasets to test generalizability. The evaluation metric (classification accuracy) is standard and appropriate for these tasks. The experimental setup, including freezing the backbone and training for 100 epochs with AdamW optimization, is reasonable and consistent with prior work (e.g., DAM-VP, InsVP). However, the choice of a single ViG-M backbone (pre-trained on ImageNet-21k) could be expanded to other ViG variants (e.g., MobileViG) to validate robustness further, though this is a minor concern given the focus on prompting efficiency.

**Other Comments Or Suggestions:**

I have no other comments or suggestions.

**Other Strengths And Weaknesses:**

**Strengths**:

*Originality*: VGP is a pioneering effort in prompting ViG models, creatively adapting low-rank concepts to graph structures.

*Significance*: The method’s parameter efficiency (94.6% reduction) and strong performance (e.g., 89.6% average accuracy) make it highly practical for resource-constrained settings.

*Clarity*: The paper is well-written, with clear explanations of the method (Section 4) and insightful visualizations (Figures 2, 3).

**Weaknesses**:

*Clarity of Generalizability*: The extension to graph tasks (Table 2) is compelling but lacks depth in explaining why low-rank properties hold beyond vision, limiting interpretability.

*Limited Backbone Variety*: Testing only on ViG-M restricts insights into broader applicability across ViG variants.
Minor Presentation Issue: The abstract could better highlight the low-rank insight as a core novelty, as it currently focuses more on the framework.

**Questions For Authors:**

**Generalizability to Graph Tasks**: Your hypothesis suggests low-rank patterns exist in chemistry/biology graph data (Section 5.3). Could you provide PCA or similar analysis on these datasets (like Figure 2 for vision) to confirm this? A positive response with evidence would strengthen my confidence in VGP’s broader applicability, potentially raising my rating from "accept" to "strong accept."

**Choice of Rank $r=32$**: Table 4 shows peak performance at $r=32$, but Appendix A.3 estimates $r=50$ (CUB) and $r=60$ (Flowers). Why was $r=32$ chosen over these values? Clarification could resolve this apparent discrepancy, impacting my view on the method’s optimization rigor.

**Relation To Broader Scientific Literature:**

The paper’s key contributions align well with trends in parameter-efficient fine-tuning (PEFT) and graph neural networks (GNNs). The use of prompting for vision tasks builds on prior work like VPT (Jia et al., 2022) and InsVP (Liu et al., 2024), extending it to ViG, a graph-based vision backbone introduced by Han et al. (2022). The low-rank decomposition idea echoes techniques in efficient Transformer adaptation (e.g., LoRA, Hu et al., 2021, not cited) but is novel in its application to graph structures. The extension to traditional graph tasks (e.g., MoleculeNet) ties into GNN prompting literature (e.g., GraphPrompt, Liu et al., 2023; GPF-Plus, Fang et al., 2023), offering a bridge between vision and graph domains. The insight into low-rank semantic properties also resonates with dimensionality reduction studies in GNNs (e.g., Kipf & Welling, 2016b), though applied uniquely to vision graphs.

**Theoretical Claims:**

The paper includes a theoretical claim in Appendix A.3 linking the low-rank property of semantic information to PCA and eigenvalue decomposition (Equations 13-15). I reviewed the correctness of this analysis, which builds on standard PCA principles to argue that semantically connected nodes share dominant components, implying a low-rank structure. The formulation appears mathematically sound: the covariance matrix decomposition and rank estimation based on eigenvalue thresholds are consistent with PCA theory. The error term estimation ($O(\lambda_{r+1})$) is a reasonable approximation, though it assumes a clear eigenvalue drop-off, which is visually supported by Figure 5’s long-tail distribution. No significant issues were found, but the analysis could be strengthened by quantifying the variance captured by the chosen rank $r$ (e.g., 50 for CUB) to directly tie it to the experimental choice of $r=32

---

> ### Author Rebuttal · Authors · 2025-03-31
>
> Thank you for your valuable feedback.
>
> ### Q1,W1. Generalizability to Graph Tasks
>
> From the efficacy of our method on chemistry/biology graph datasets, we hypothesize **similar latent semantic low-rank patterns** also exist in these graph data. In particular, **chemical bonds** and **protein interaction** structures exhibit structured low-rank properties similar to **semantic regions in images**.
>
> For example, in *PPI* (Protein-Protein Interaction) dataset, each node in graph represents a type of protein, while edges denote interaction relationships. These interactions are primarily driven by **specific functional groups** such as hydroxyl and carboxyl groups, which are crucial to biochemical reactions, analogous to **low-rank semantic features in vision images**. Conversely, other chemical groups that do not significantly contribute to interactions correspond to **high-frequency local details** in images, which tend to be redundant.
>
> So when extracting features from these protein graph data for tasks such as protein function prediction, it is essential to **identify the key functional groups** that drive interactions. Since our **VGP model is designed to capture low-rank semantic structures**, it effectively generalizes to chemistry and biology graph datasets, surpassing prior graph prompting methods.
>
> Due to the inherent abstract nature of protein interaction graphs, it is challenging to visualize this similar semantic pattern like 2D images in Figure 2. To illustrate the underlying patterns, we visualize the graph structures and color the nodes with corresponding the first three PCA components. We make comparisons between node features obtained from trained and untrained GNN models on *PPI* dataset. Interestingly, we find that node features from trained GNN models present significantly better low-rank features consistency. The visualization is provided in an anonymous link(https://anonymous.4open.science/r/ICML25-anonymous-DF1B/PPI-PCA.pdf).
>
> ### Q2. Choice of Rank $r=32$
>
> Our choice of $r=32$ is motivated by two key considerations:
>
> 1) **Trade-off between performance and parameter efficiency**. As shown in Table 4, *CUB* gets near-optimal results with **87.4%** of $r=32$ and **87.2%** of $r=64$ and its estimated rank is about 50, falling between these two values. Besides, *CIFAR* gets peak results when $r=64$ and second-best performance when $r=32$. In aspect of performance, choose $r=32$ or $r=64$ both seems plausible. However, in aspect of parameter efficiency, increasing $r$ from 32 to 64 **nearly doubles the trainable parameters**. So we tend to choose the smaller one, as $r=32$ for an optimal balance between performance and parameter efficiency.
> 2) **Consistent hyperparameters across datasets**. To maintain a unified hyperparameter setting and avoid dataset-specific tuning, we adopt consistent $r=32$ across all the datasets. Even though estimated ranks of *CUB* and *Flowers* are 50 and 60 respectively, other datasets like *SVHN* and *CIFAR10* exhibit lower ranks as 18 and 20, as table shown below. To reach a reasonable compromise between these datasets, we set $r=32$ to satisfy majority of datasets.
>
> |               | *DTD* | *CUB* | *NABirds* | *Dogs* | *Flowers* | *Food* | *CIFAR* | *CIFAR10* | *GTSRB* | *SVHN* |
> | :-----------: | :---: | :---: | :-------: | :----: | :-------: | :----: | :-----: | :-------: | :-----: | :----: |
> | estimated $r$ |  36   |  50   |    90     |   30   |    60     |   46   |   55    |    20     |   26    |   18   |
>
>
>
> ### W2. Backbone Variety
>
> We supplement additional experiments based on other representative graph-based vision models, including MobileViG and GreedyViG. The experiments are conducted on six vision datasets and the backbones are pre-trained on ImageNet-1k. As table shown below, our VGP consistently excels other SOTA vision prompting and graph prompting methods, demonstrating robustness across backbones due to our adaptability to diverse graph structures.
>
> |  Method  |  *DTD*   |  *CUB*   | *Flowers* |  *Food*  | *CIFAR10* |  *SVHN*  | Average  |
> | :------: | :------: | :------: | :-------: | :------: | :-------: | :------: | :------: |
> |          |          |          | MobileViG |          |           |          |          |
> | GPF-Plus |   68.5   |   81.4   |   94.3    |   82.0   |   94.1    |   82.2   |   83.7   |
> |  InsVP   |   68.1   |   84.0   |   95.2    |   83.9   |   95.2    |   88.9   |   85.9   |
> | **VGP**  | **71.6** | **84.9** | **97.6**  | **88.0** | **96.8**  | **94.7** | **88.9** |
> |          |          |          | GreedyViG |          |           |          |          |
> | GPF-Plus |   69.3   |   81.7   |   94.9    |   82.2   |   94.5    |   82.7   |   84.2   |
> |  InsVP   |   68.8   |   84.1   |   95.5    |   84.0   |   95.5    |   89.1   |   86.2   |
> | **VGP**  | **72.1** | **85.4** | **98.0**  | **87.3** | **97.2**  | **94.5** | **89.1** |

---

### Official Review · Reviewer_WsNv · 2025-03-09

**Overall Recommendation:** 3

**Summary:**

This paper introduces a novel parameter-efficient method called **V**ision **G**raph **P**rompting (**VGP**) with semantic low-rank decomposition for Vision GNNs. Empirical results demonstrate that the proposed approach achieves impressive performance on both image classification and traditional graph classification tasks. In addition, the paper provides supporting visualization evidence via PCA, which underscores the motivation behind the low-rank decomposition design in the prompts.

**Claims And Evidence:**

The authors assert that semantically connected components in the graph exhibit low-rank properties, as evidenced by visualizations produced using PCA and t-SNE. However, I believe this visualization approach may have a critical limitation due to the capacity of PCA in effectively extracting the target object in complex images. Specifically, PCA may struggle to isolate the target object when selecting the top components, particularly in scenarios where images contain multiple objects or intricate backgrounds. Therefore, I suggest providing additional visualization results of PCA components, especially for images with multiple objects and complex backgrounds, to better evaluate the method’s effectiveness under such conditions.

**Essential References Not Discussed:**

I am not an expert in this field, so I am unsure if there are any related references that have not been cited in this paper.

**Experimental Designs Or Analyses:**

The experimental results of the proposed method, as presented in Tables 1 and 2, are promising. However, the authors do not appear to provide sufficient analysis regarding the differences between ViT-based methods and ViG-based ones.

Firstly, the parameter sizes of the selected backbones remain unclear and should be explicitly stated for better comparison. Secondly, it is recommended that the authors analyze why the basic visual prompting method for ViTs (i.e., VPT) outperforms ViG-based prompting methods on certain datasets, such as CUB, NABirds, Dogs, and Flowers. Specifically, the authors should offer a more detailed discussion on the advantages and disadvantages of ViT-based and ViG-based prompting methods. This would help readers better understand the critical discrepancies between these two approaches.

**Methods And Evaluation Criteria:**

Overall, the proposed method, VGP, provides an effective solution for leveraging semantic graph information within a low-rank space. Specifically, the semantic low-rank decomposition framework of VGP, including the SeLo-Graph Prompt, SeLo-Edge Prompt, and SeLo-Node Prompt, facilitates both structural adaptation and feature enhancement.

**Other Comments Or Suggestions:**

I have no further comments.

**Other Strengths And Weaknesses:**

Strengths:

1. The originality of the proposed method stems from the innovative combination of existing ideas, including low-rank adaptation and Vision GNN, which demonstrates a creative and thoughtful approach.

2. The writing in this paper is clear and well-structured, making it easy for readers to understand both the motivation behind the work and the effectiveness of the proposed method.

Weaknesses:

My major concerns can be found in previous parts of Claims And Evidence, Theoretical Claims, and Experimental Designs Or Analyses.

**Questions For Authors:**

I am curious about the cluster located in the bottom-right corner of Figure 1. It appears to differ significantly from the other clusters. While the patches in the top figure seem to represent the background of the overall image, it is unclear what this particular cluster corresponds to. Could you clarify its meaning or significance?

**Relation To Broader Scientific Literature:**

The authors validate the effectiveness of the proposed method solely on image classification and graph classification tasks in this paper. However, it remains unclear whether the method can be extended to other vision tasks, such as object detection and segmentation.

**Theoretical Claims:**

I have carefully reviewed the theoretical aspects of this paper and did not identify any obvious errors. However, I noticed that the authors did not provide an equation summarizing the proposed three modules for parameter updating. Including such an equation is recommended to enhance the clarity and understanding of the overall method.

---

> ### Author Rebuttal · Authors · 2025-03-31
>
> Thank you for your valuable feedback.
>
> ### Q1. Cluster Located in Bottom-Right Corner of Figure 1
>
> We further check the correspondence between the t-SNE clusters and images patches, finding that the bottom-right cluster corresponds to **the bird's reflection on the water**. This observation aligns with the PCA visualizations in Figures 1 and 2, where the bird’s reflection is also highlighted.
>
> Interestingly, the ViG model appears to learn semantic information about reflections as a byproduct of supervision in the bird classification task.
>
> However, since the model lacks explicit supervision on background elements, the background features exhibit a sparse distribution in the upper region of the t-SNE figure.
>
> ### W1. PCA with Multiple Objects and Complex Backgrounds
>
> In Figure 2(b) of our paper, the samples from *Flowers* dataset already contains **complex backgrounds with cluttered grass and leaves**, as well as instances with multiple objects (e.g., **the top-right sample with two flowers**).
>
> The results demonstrate that PCA effectively extracts target objects using trained ViG model’s features, attributed to the **semantic low-rank property** of the ViG's latent feature space. In the final version, we will provide additional visualizations specifically focusing on multiple objects and complex backgrounds.(https://anonymous.4open.science/r/ICML25-anonymous-DF1B/multi-objects-w-complex-backgrounds.pdf)
>
> ### W2. Summarizing Proposed Three Modules for Training
>
> Combining Equation 11 and 12 in the paper, we further summarize the prompted ViG model as equation below. The updated modules during training are underlined. Only the three **low-rank prompt matrices**, one **semantic feature extraction MLP** and the **low-rank virtual nodes** are trained, while all other modules in the ViG backbone remain frozen:
>
> $\hat{f}(\mathbf{x}\_i)= (1-\beta)\cdot\mathbf{x}\_i+\hat{g}(\mathbf{x}\_i, \underline{\mathbf{P\_n}}) \cdot \mathbf{W}\_{update} +\sum\_{\mathbf{x}\_j \in \hat{\mathcal{N}}(\mathbf{x}\_i)} \beta\cdot  \underline{\mathrm{MLP\_s}}(\mathbf{x}\_j)\cdot\underline{\mathbf{P\_e}}, ~~~\hat{\mathcal{N}}(\mathbf{x}\_i)  \subseteq [\mathbf{X},~[\underline{\mathbf{n}\_1, \dots, \mathbf{n}\_M}]\cdot \underline{\mathbf{P\_g}}]$
>
> ### W3. Differences between ViT-based and ViG-based Prompts
>
> The ViT-based prompting methods either prompting on images pixels like VP, explicitly adjusting the RGB channels space, or prompting on image tokens like VPT, functioning via feature similarity-based attention mechanism.
>
> However, these methods lack awareness of **graph structures**, such as edge connections between patches. While ViG-based graph prompting methods like GraphPrompt and GPF-Plus explicitly alters graph structures, including modifying node features, inserting new nodes and constructing new edges, thus better leveraging the graph representation.
>
> As for ViT-based prompting methods outperform ViG-based one on certain datasets, this is likely because vision datasets contain both visual features and latent graph structures. ViT-based vision prompting methods excel in processing **raw vision data**, whereas ViG-based methods are more effective at **graph-based reasoning**. Consequently, each approach has its own advantages, leading to instances where ViT-based prompting achieves superior results.
>
> ### W4. Parameter Sizes of the Backbone
>
> We have provided details of both parameter sizes and computation costs of backbone ViG and our VGP in **Appendix A.1 and Table 5**. The ViG model has **48.68M** parameters averagely and our VGP only has **2.61M** trainable parameters, reducing **94.6%** from full fine-tuning.
>
> ### W5. Extending to Other Vision Tasks
>
> We supplement additional semantic segmentation tasks on *ADE20K* dataset. As table shown below, our method consistently excels other vision and graph prompting methods on semantic segmentation tasks, demonstrating its effectiveness across different vision tasks.
>
> |Method|ViG-M|Adapter|VPT|InsVP|GraphPrompt|VGP|
> |:------:|:---:|:-----:|:--:|:---:|:---------:|:------:|
> |mIoU(%)|47.9|44.2|41.6|42.3|44.4|**47.6**|

---

### Official Review · Reviewer_kLEh · 2025-03-11

**Overall Recommendation:** 4

**Summary:**

This paper proposes a novel approach called Vision Graph Prompting (VGP), which enables parameter-efficient fine-tuning of the Vision GNN (ViG) model. Additionally, the paper observes that essential semantic information in Vision Graph structures is concentrated in low-rank components and leverages this insight to introduce a Semantic Low-Rank Decomposition-based prompting method. To capture both global and local semantic features within the graph structure, three key components—SeLo-Graph, SeLo-Edge, and SeLo-Node Prompt—are introduced. Experimental results demonstrate that this approach significantly enhances the transfer learning performance of the ViG model while requiring far fewer parameters compared to full fine-tuning.

**Claims And Evidence:**

- This paper visually demonstrates, through Figure 2 and Figure 5, that the primary semantic information of the Vision Graph is concentrated in the lower-dimensional components via PCA analysis. Additionally, the Ablation Study in Table 3 experimentally proves that SeLo-Graph, SeLo-Edge, and SeLo-Node Prompt each contribute to performance improvement.
- The experimental results in Table 1 and Table 2 further indicate that the proposed method outperforms existing approaches across various benchmark experiments.

- However, there is a lack of experiments comparing the cases with and without the application of low-dimensional decomposition. Therefore, it remains unclear how significant the performance improvement is compared to the basic ViG.
- It is necessary to provide a more detailed explanation of why the graph prompting technique is structurally optimized for the ViG model.
- In this paper, the impact of blending factors α and β on performance is discussed, indicating that the optimal values are found within a specific range (0.1 to 0.3). Although this claim is supported by quantitative analysis, more detailed insights should be provided regarding why deviations from this range lead to performance degradation.

**Essential References Not Discussed:**

- The paper effectively summarizes existing research, particularly related to Vision GNN and Vision Prompting, and does not appear to have omitted any essential studies that should have been mentioned.

**Experimental Designs Or Analyses:**

- This paper has appropriately set up comparison groups for the experiments and conducted a comprehensive comparative analysis, including existing prompting techniques (VPT, InsVP) and the graph prompting (GPF-Plus) technique. The Ablation Study verifies the contribution of each proposed technique to performance improvement, ensuring logical validity.


- However, providing more explicit information on the number of repetitions (epoch) would enhance the transparency of the experimental design.
- This paper claims significant improvements in transfer performance, but it does not include detailed statistical analyses or significance testing results. Incorporating such analyses would help assess the robustness of the findings and is essential for demonstrating that the observed improvements are not merely due to random chance.

**Methods And Evaluation Criteria:**

- This paper conducts experiments using various benchmarks (CIFAR, CUB, GTSRB, and chemical/biological graph data) and appropriately analyzes the contribution of each technique through an Ablation Study. Additionally, the evaluation considering the balance between parameter efficiency and performance appears to be well-justified.

- While the paper claims that the proposed method achieves results comparable to full fine-tuning, it does not specify the exact performance evaluation metrics used for comparison. Including metrics such as accuracy, F1 score, or AUC would provide clearer insights into the effectiveness of the proposed method.
- Additional explanations on the graph datasets should be provided. While it appears that datasets from GPF-PLUS and MoleculeNet were used, there is a lack of detailed descriptions regarding their characteristics (even the appendix does not provide an explanation).
- It is necessary to verify whether the proposed method demonstrates the same effectiveness in other graph-based vision models (e.g., MobileViG, GreedyViG).
Further analysis should be conducted to determine whether the performance of the proposed low-dimensional decomposition method varies depending on the dataset.
- A clearer analysis of how changes in graph structure affect performance during the prompting process would be beneficial.

**Other Comments Or Suggestions:**

- Grammar errors and inaccurate expressions:
"textbf3.1%" → "3.1%" (Typo correction needed)
"demonstrated in Table reftab-datasets" → "demonstrated in Table 6" (Reference error correction)
- It is necessary to more clearly articulate the limitations of the study in the conclusion.
For example: analysis of the causes of poor performance on certain datasets, suggesting additional research directions, etc.
- Providing a clearer explanation of the PCA visualization in Figure 5 would be beneficial.
The current explanation is somewhat brief, which may make it difficult for readers to easily understand the meaning of the graph.

**Other Strengths And Weaknesses:**

- The paper proposes the first Vision Graph Prompting technique for the ViG model, demonstrating high originality. It achieves superior performance compared to existing methods while maintaining parameter efficiency. The inclusion of an Ablation Study and various benchmark experiments enhances the reliability of the research.

- It would be beneficial to include additional statistical significance for the experimental results. There are changes in model performance based on hyperparameters (r, α, β), and it may be necessary to include a process for finding the optimal values.

**Questions For Authors:**

- Will the proposed method show the same effect in other graph-based vision models (e.g., MobileViG, GreedyViG)?
- How will performance change if the structure of the graph is altered?
- Has the impact of prompting on the model's explainability been analyzed?
- Is it possible to achieve the same performance improvements in other domains, such as autonomous driving, medical imaging, and remote sensing?

**Relation To Broader Scientific Literature:**

- The paper effectively summarizes the relationship with existing Vision GNN and Vision Prompting research, clearly explaining the differences from Transformer-based prompting techniques

- It is necessary to verify whether the proposed method demonstrates the same effectiveness in other graph-based vision models (e.g., MobileViG, GreedyViG).

**Theoretical Claims:**

- This paper demonstrates through PCA-based analysis that the semantic information of the Vision Graph is primarily contained in low-dimensional components. By utilizing Eigenvalue Decomposition (EVD) of the Covariance Matrix, it shows that semantic information is concentrated in a few principal components. Furthermore, based on this mathematical foundation, it logically validates the effectiveness of Semantic Low-Rank Decomposition.

- This paper claims that the proposed method effectively captures critical semantic information in Vision GNNs, thereby enhancing feature extraction. While this claim is supported by experimental results, the theoretical justification for how this improvement is achieved through low-rank decomposition and graph adaptation needs to be clearly articulated.
- The paper explains that extensive experiments demonstrate significant improvements in transfer performance across various downstream tasks. However, the logical connection between the theoretical claims and the experimental results should be further strengthened. In particular, providing a more detailed explanation of how the proposed theoretical framework translates into practical performance gains would enhance the coherence of the paper and establish a clearer pathway from theory to application.

---

> ### Author Rebuttal · Authors · 2025-03-31
>
> Thank you for your valuable feedback.
>
> ### Q1. Experiments on Other Graph-based Vision Models
> We supplement additional experiments on other graph-based vision models, including **MobileViG and GreedyViG**, across six vision datasets with ImageNet-1k pre-trained backbones. As table shown below, our VGP consistently excels other SOTA vision and graph prompting methods, demonstrating robustness across backbones due to our adaptability to diverse graph structures.
>
> |Method|*DTD*|*CUB*|*Flowers*|*Food*|*CIFAR10*|*SVHN*|Average|
> |:------:|:---:|:---:|:-------:|:----:|:-------:|:----:|:------:|
> ||||MobileViG|||||
> |GPF-Plus|68.5|81.4|94.3|82.0|94.1|82.2|83.7|
> |InsVP|68.1|84.0|95.2|83.9|95.2|88.9|85.9|
> |**VGP**|71.6|84.9|97.6|88.0|96.8|94.7|**88.9**|
> ||||GreedyViG|||||
> |GPF-Plus|69.3|81.7|94.9|82.2|94.5|82.7|84.2|
> |InsVP|68.8|84.1|95.5|84.0|95.5|89.1|86.2|
> |**VGP**|72.1|85.4|98.0|87.3|97.2|94.5|**89.1**|
>
> ### Q2. Ablation on Altered Graph Structures
> We conduct additional ablation studies to analyze the impact of structural modifications in the SeLo-Graph Prompt. Our method **inserts virtual nodes** and **dynamically constructs edges** based on feature similarity, thereby altering the graph structure.
>
> As table shown below, both virtual node insertion and edge construction enhance feature extraction. While static edge allocation improves performance, **dynamic edge construction based on feature similarity achieves the best results**, as it better captures complex semantic relationships.
>
> |**Ablation**|**CUB**|**GTSRB**|
> |------------------------------------------------|--------|---------|
> |w/o SeLo-Graph Prompt|85.8|93.4|
> |Only insert virtual nodes|86.2|94.1|
> |Insert virtual nodes+static edge allocation|86.6|96.9|
> |Insert virtual nodes+dynamic edge construction|**87.4**|**98.1**|
>
> ### Q3. Analysis of Explainability of Prompting Impact
> Fig.2 and Fig.5 in paper show that fully fine-tuned vision graph models can recognize semantically related patches and connect them via edges, exhibiting a semantic low-rank property.
>
> Our VGP reinforces this effect through three prompting modules, ensuring **low-rank feature consistency** across connected patches. This mimics the behavior of fully fine-tuned models, effectively linking semantically related regions.
>
> As shown in Table 1, our method effectively extracts discriminative semantic information, leading to significant performance gains.
>
> ### Q4. Experiments in Other Domains
> We supplement additional experiments on **remote sensing tasks on *EuroSAT* dataset**, which consists of satellite images from Sentinel-2. As table shown below, our method still compels other SOTA prompting methods, even getting comparable results with full fine-tuning with only **4%** trainable parameters.
>
> |Method|ViG-M|Adapter|VPT|InsVP|GraphPrompt|GPF-Plus|VGP|
> |:----------------:|:---:|:-----:|:---:|:---:|:---------:|:------:|:-------:|
> |*EuroSAT* Acc.(%)|92.37|85.24|83.55|87.14|85.50|86.97|**91.98**|
>
> ### W1. Statistical Significance and Hyperparameter Selection
> We run all experiments for **three times** with different seeds and report the highest results. The average standard deviation is **0.3%**, which is significant lower than our **4%** performance gain, confirming the robustness of our results. Detailed statistical significance for each dataset will be provided in final version.
>
> For hyperparameter selection (r, α, β), we evaluate multiple candidates and select the optimal ones. And we use a fixed set of hyperparameters across all datasets.
>
> ### S1,S2,S3. Comments and Suggestions
> We appreciate your feedback and will address the following in final version:
>
> 1) We have corrected typos in the supplementary materials.
> 2) We will discuss limitations and failure cases to guide future research.
> 3) Additional PCA visualization details are included. Specifically, we compute PCA components for all patch tokens encoded by the trained model. The latent feature space is decomposed into PCA components, where those with large coefficients capture major variance. We map the top three PCA components into RGB channels for visualization, ensuring patches with similar colors share similar PCA component distributions, thus indicating low-rank properties. This approach is similar to visualization of famous DINOv2.
>
> ### Graph Prompt Structurally Optimized for ViG
> Standard vision prompting methods (e.g., VP, VPT) operate on **pixel-level** or **token-level** representations without explicit graph structures.
>
> In contrast, graph prompting methods (e.g., GraphPrompt, GPF) **directly modify node features, insert virtual nodes, or establish new edges**, enabling structured graph-based prompting while overlooking semantic features in vision data.
>
> Our VGP builds upon this principle, explicitly optimizing prompts for graph-structured vision models, incorporating semantic low-rank decomposition strategy.
>
> ### Number of Repetitions
> Following DAM-VP (Appendix A.2), we train each dataset for **100 epochs**.

---

> > ### Comment · Reviewer_kLEh · 2025-04-02
> >
> > The authors have provided sincere and well-reasoned responses to all of the reviewer’s questions. In particular, they effectively demonstrated the generalizability and extensibility of the proposed method through additional experiments on alternative backbones (MobileViG, GreedyViG) and the remote sensing domain (EuroSAT). They also convincingly explained the effects of graph structure modifications and the improvement in explainability brought by the prompting technique, supported by both quantitative and qualitative evidence. Plans to supplement statistical significance analysis and hyperparameter selection were clearly stated as well.
> >
> > However, there are a few shortcomings. First, regarding statistical validation, conducting only three runs and reporting only the best performance may be somewhat insufficient in terms of consistency and reliability. In addition, among the various domain experiments, real-world applications such as autonomous driving or medical imaging—which are more complex—were not tested. Furthermore, the paper focuses more on overall system performance improvement rather than providing in-depth analysis on the causes of individual performance gains, which weakens the connection between theory and experiments. Addressing these issues in the future would further enhance the completeness of the paper.
> >
> > Taking all these points into account, I will adjust my previous score slightly upward. I wish the authors the best with their final results.

---

> > > ### Author Response · Authors · 2025-04-07
> > >
> > > Thank you very much for your thorough and constructive feedback. We sincerely appreciate your recognition of our efforts to address your concerns, especially regarding the generalizability and explainability of our method.
> > >
> > > We acknowledge the limitations you raised. Regarding statistical validation, we agree that more comprehensive experimentation (e.g., more runs with mean and standard deviation) would enhance the reliability of our results, and we plan to incorporate this in future work. We also appreciate your suggestion on exploring more complex real-world domains such as autonomous driving and medical imaging—this is a valuable direction that we are actively considering for follow-up research.
> > >
> > > Lastly, we agree that a deeper analysis into the individual contributions of each module would strengthen the theoretical-experimental connection, and we will work toward expanding this aspect in a future extended version of the paper.
> > >
> > > Thank you again for your thoughtful comments and for adjusting your score.

---

### Official Review · Reviewer_x6eb · 2025-03-15

**Overall Recommendation:** 3

**Summary:**

In this work, the authors present Vision Graph Prompting (VGP), a parameter-efficient fine-tuning method for Vision Graph Neural Networks. The core insight is that semantic information in vision graphs primarily resides in the low-rank components of the latent feature space. The authors propose three semantic low-rank prompting methods: SeLo-Graph, SeLo-Edge, and SeLo-Node prompts, which capture global structural patterns and fine-grained semantic dependencies.

**Claims And Evidence:**

Overall, the claims are well-supported and clear.

**Essential References Not Discussed:**

N/A.

**Ethical Review Concerns:**

N/A.

**Experimental Designs Or Analyses:**

The ablation study in the paper does not include experiments where only SeLo-Edge or only SeLo-Node is used, nor does it show the results of combining SeLo-Graph with SeLo-Node. This limits the thoroughness of the analysis of each component's individual contribution.

**Methods And Evaluation Criteria:**

The evaluation criteria, including accuracy and parameter efficiency across diverse datasets, are appropriate and comprehensive.

**Other Comments Or Suggestions:**

N/A.

**Other Strengths And Weaknesses:**

Strengths:
1. The authors present a parameter-efficient fine-tuning method specifically designed for Vision Graph Neural Networks (ViG), addressing a previously under-explored area.

2. The core insight regarding the low-rank properties of semantic information in vision graphs is well-supported and motivated.

3. Extensive experiments across diverse datasets demonstrate the effectiveness of the proposed method.

Weaknesses:

1. Missing detailed information on the implementation of virtual nodes, such as their initialization method and how the number of virtual nodes is determined.

2. The author does not discuss potential issues that may arise when dealing with very large or complex graph structures. Additionally, the author does not clarify the number of prompts M in the SeLo-Graph Prompt, which may also be crucial to the the effectiveness of the method.

3. The proposed method appears to be instance-level, which may result in a significantly larger number of parameters compared to more generic prompt methods. Though a comparison with the full fine-tuning method is given in the appendix, the paper does not provide a detailed comparison of parameter quantities, which is important for assessing the method's efficiency.

4. It is unclear whether SeLo-Node acts on all nodes in the graph or if it includes the virtual nodes from SeLo-Graph. This ambiguity affects the understanding of the method's application scope and its impact on different parts of the graph.

**Questions For Authors:**

I'm curious, given the authors' finding that the original graph structure's features have a low-rank decomposition property, could they consider adding LoRA to ViG for fine-tuning as a prompt alternative?

**Relation To Broader Scientific Literature:**

This work bridges the gap between Transformer-focused prompting techniques and graph-based vision models, contributing to the development of parameter-efficient fine-tuning methods for ViG and potentially other graph neural network applications.

**Theoretical Claims:**

I roughly checked the theoretical claims in the article. They're mostly based on existing theories and seem reasonable.

---

> ### Author Rebuttal · Authors · 2025-03-31
>
> Thank you for your valuable feedback.
>
> ### Q1. Adding LoRA as a Prompt Alternative
> We supplement additional experiments comparing with LoRA across ten vision datasets. As table shown below, LoRA surpasses traditional visual prompting method (VPT) due to its **low-rank adaptation property** and matches GPF-Plus’s performance.
>
> While LoRA focuses on low-rank adaptation within the model’s parameter space, it **does not leverage graph topology**, unable to refine structural relationships, limiting further gains. Our VGP achieves SOTA performance by jointly optimizing visual semantics and graph structures via semantic low-rank prompting.
>
> |Method|*DTD*|*CUB*|*NABirds*|*Dogs*|*Flowers*|*Food*|*CIFAR*|*CIFAR10*|*GTSRB*|*SVHN*|Average|
> |:------:|:------:|:------:|:-------:|:------:|:-------:|:------:|:------:|:-------:|:------:|:------:|:------:|
> |VPT|71.4|77.3|76.4|73.1|95.3|81.9|76.3|93.2|79.7|82.4|80.7|
> |GPF-Plus|71.0|82.0|77.2|78.2|95.7|82.6|80.9|94.5|90.5|83.1|83.6|
> |LoRA|69.7|79.2|77.3|74.0|94.6|83.5|81.4|94.9|90.2|90.8|83.6|
> |**VGP**|**74.8**|**87.4**|**80.9**|**81.7**|**98.2**|**89.5**|**89.7**|**98.3**|**98.1**|**96.9**|**89.6**|
>
> ### W1. Implementation Details of Virtual Nodes
> Thanks for your reminder. We provide additional implementation details of virtual nodes:
>
> 1) The virtual nodes in SeLo-Graph Prompt are initialized using **Kaiming Normal distribution**.
>
> 2) The number of virtual nodes $M$ is set to **14**. Ablation experiments is conducted as table below. A smaller number leads to suboptimal prompting effects due to insufficient guidance, while an excessive number does not yield further improvements but incurring additional parameter cost.
>
> |Virtual Node Number $M$|0|3|7|14|28|42|
> |:---------------------:|:--:|:--:|:--:|:------:|:--:|:--:|
> |*CUB*|85.8|86.3|86.9|**87.4**|87.2|86.9|
> |*GTSRB*|93.4|95.5|97.3|**98.1**|98.0|97.6|
> |*SVHN*|95.1|96.0|96.2|**96.9**|96.7|96.8|
>
> ### W2. Dealing with Large or Complex Graph
> Our VGP is capable of handling large and complex graph data. In our experiments on chemistry/biology graph datasets in Table 2, the number of nodes can reach **5,000** with **non-uniform edges distribution**, far more complex than the **196-nodes** image graphs. The table below presents the average graph sizes for different chemistry/biology datasets. Even though, our VGP still achieves **seven** SOTA results across nine benchmarks with only **0.15M** parameters, verifying its robustness and generalizability.
>
> As for number of prompts $M$ in the SeLo-Graph Prompt, we follow the same setting with vision datasets as 14, not specifically tuned for chemistry/biology datasets. Even though the graph are much larger and more complex graphs, our method still excels with other graph prompting methods with a general hyperparameter setting, verifying its robustness.
>
> |Datasets|*BBBP*|*Tox21*|*ToxCast*|*SIDER*|*ClinTox*|*MUV*|*HIV*|*BACE*|*PPI*|
> |:--------:|:----:|:-----:|:-------:|:-----:|:-------:|:---:|:---:|:----:|:------:|
> |Graph Size|776|516|583|741|989|828|893|1074|**5139**|
>
> ### W3. Parameter Quantities Comparison
> We provide a parameter comparison of SOTA methods on the *CUB* dataset, as table shown below. Our method achieves high efficiency, requiring only **2.63M** trainable parameters (**5%** of ViG-M’s full fine-tuning at **48.71M**). This is due to our **lightweight low-rank design**, which avoids large parameter matrices while maintaining strong performance.
>
> |Method|ViG-M|VPT|Ins-VP|GPF-Plus|Adapter|DAM-VP|VGP|
> |:--------:|:---:|:--:|:----:|:------:|:-----:|:----:|:--:|
> |Param.(M)|48.71|1.77|1.83|2.29|3.48|6.24|2.63|
>
> ### W4. Whether SeLo-Node Prompt Acts on Virtual Nodes
> Yes, the SeLo-Node Prompt acts on all the nodes within graph, including virtual nodes inserted by SeLo-Graph Prompt. We will provide a more explicit description of prompting process in final version for better clarity as below.
>
> 1) SeLo-Graph Prompt inserts virtual nodes and builds virtual edges, updating graph structures
> 2) SeLo-Edge Prompt refining the edge-level semantic interactions via edges within updated graph
> 3) SeLo-Node Prompt intensifies node-level semantic information on each node in updated graph
>
> ### Each Component's Individual Contribution
> We supplement additional ablation experiments on different components combinations as table shown below. While SeLo-Graph Prompt refines graph structures, SeLo-Edge and SeLo-Node Prompt enhance low-rank semantics between and within nodes. Each component contributes to performance gains.
>
> |SeLo-Graph|SeLo-Edge|SeLo-Node|*CUB*|*GTSRB*|
> |:--------:|:-------:|:-------:|:------:|:------:|
> |-|-|-|76.2|77.4|
> |√|-|-|81.9|86.9|
> |-|√|-|82.3|87.5|
> |-|-|√|81.0|86.5|
> |√|√|-|85.3|93.0|
> |√|-|√|85.5|93.3|
> |-|√|√|85.8|93.4|
> |√|√|√|**87.4**|**98.1**|

---

> > ### Comment · Reviewer_x6eb · 2025-04-05
> >
> > Thanks for the authors' response. Since most of my concerns have been addressed, I am inclined to increase my score.

---

> > > ### Author Response · Authors · 2025-04-07
> > >
> > > Thanks for your positive feedback and for considering increasing your score. We truly appreciate your thoughtful review and are glad that our rebuttal addressed your concerns. We are committed to improving our work and are grateful for your constructive comments, which helped us strengthen the paper.

---

### Decision · Program_Chairs · 2025-05-01

**Decision:**

Accept (poster)

**Comment:**

This paper received four positive ratings, with all reviewers generally inclined to accept it. The paper presents Vision Graph Prompting (VGP), a novel parameter-efficient fine-tuning method tailored for Vision Graph Neural Networks (ViG). Different from the prompting methods designed for transforms, the author specifically considers the rich topological relationships between nodes and edges in the visual graph structure, which improves the modeling ability of complex semantics. According to the reviews, this paper is well written, with a well-explained methodology, clear motivation and insightful visualizations that aid understanding. All reviewers recognized the novelty of the proposed method, which innovatively explores efficient parameter fine-tuning for the ViG model. In addition, the authors conduct extensive experiments across different datasets and with different backbone networks to effectively demonstrate the generality and scalability of the proposed method.
The authors have addressed the concerns raised, resolving most of the doubts. Therefore, the Area Chair (AC) recommends accepting the paper.